# Efficacy of FFP3 respirators for prevention of SARS-CoV-2 infection in healthcare workers

Mark Ferris[1,2]*[†], Rebecca Ferris[1†], Chris Workman[1], Eoin O'Connor[3], David A Enoch[1,4], Emma Goldesgeyme[1], Natalie Quinnell[1], Parth Patel[1], Jo Wright[1], Geraldine Martell[1], Christine Moody[1], Ashley Shaw[1], Christopher JR Illingworth[5,6,7‡], Nicholas J Matheson[1,8,9,10‡], Michael P Weekes[1,8,11]*[‡]

[1]Cambridge University Hospitals NHS Foundation Trust, Cambridge, United Kingdom; [2]University of Cambridge Occupational Health and Safety Service, Cambridge, United Kingdom; [3]School of Clinical Medicine, Cambridge, United Kingdom; [4]Clinical Microbiology & Public Health Laboratory, Public Health England, Cambridge, United Kingdom; [5]MRC Biostatistics Unit, Cambridge, United Kingdom; [6]Department of Applied Mathematics and Theoretical Physics, Cambridge, United Kingdom; [7]MRC-University of Glasgow Centre for Virus Research, Scotland, United Kingdom; [8]Department of Medicine, University of Cambridge, Cambridge, United Kingdom; [9]Cambridge Institute of Therapeutic Immunology and Infectious Disease (CITIID), Jeffrey Cheah Biomedical Centre, Cambridge, United Kingdom; [10]NHS Blood and Transplant, Cambridge, United Kingdom; [11]Cambridge Institute for Medical Research, Cambridge, United Kingdom

*For correspondence:
mrmf2@cam.ac.uk (MF);
mpw1001@cam.ac.uk (MPW)

[†]These authors contributed equally to this work
[‡]These authors also contributed equally to this work

Competing interest: The authors declare that no competing interests exist.

## Abstract

**Background:** Respiratory protective equipment recommended in the UK for healthcare workers (HCWs) caring for patients with COVID-19 comprises a fluid-resistant surgical mask (FRSM), except in the context of aerosol generating procedures (AGPs). We previously demonstrated frequent pauci- and asymptomatic severe acute respiratory syndrome coronavirus 2 infection HCWs during the first wave of the COVID-19 pandemic in the UK, using a comprehensive PCR-based HCW screening programme (Rivett et al., 2020; Jones et al., 2020).

**Methods:** Here, we use observational data and mathematical modelling to analyse infection rates amongst HCWs working on 'red' (coronavirus disease 2019, COVID-19) and 'green' (non-COVID-19) wards during the second wave of the pandemic, before and after the substitution of filtering face piece 3 (FFP3) respirators for FRSMs.

**Results:** Whilst using FRSMs, HCWs working on red wards faced an approximately 31-fold (and at least fivefold) increased risk of direct, ward-based infection. Conversely, after changing to FFP3 respirators, this risk was significantly reduced (52–100% protection).

**Conclusions:** FFP3 respirators may therefore provide more effective protection than FRSMs for HCWs caring for patients with COVID-19, whether or not AGPs are undertaken.

**Funding:** Wellcome Trust, Medical Research Council, Addenbrooke's Charitable Trust, NIHR Cambridge Biomedical Research Centre, NHS Blood and Transfusion, UKRI.

## Editor's evaluation

Respiratory protective equipment that is recommended in the UK for health-care workers caring for COVID-19 patients comprises a fluid resistant surgical mask (FRSM), and in case of procedures that generate aerosols FFP3 respirators are to be used. In this study, health-care workers using FRSMs, while working on COVID-19 wards faced an approximately 31-fold increased risk of ward-based SARS CoV-2 infection. After changing to FFP3 respirators, this risk was significantly reduced. Thus, FFP3 respirators seem to provide more protection than FRSMs for health-care workers caring for patients with COVID-19.

## Introduction

Consistent with World Health Organization (WHO) advice (*World Health Organization, 2021*), UK Infection Protection Control guidance recommends that healthcare workers (HCWs) caring for patients with coronavirus disease 2019 (COVID-19) should use fluid-resistant surgical masks (FRSMs) type IIR as respiratory protective equipment (RPE), unless aerosol generating procedures (AGPs) are being undertaken or are likely, when a filtering face piece 3 (FFP3) respirator should be used (*UK Government, 2021a*). Following a recent update, an FFP3 respirator is now also recommended if 'an unacceptable risk of transmission remains following rigorous application of the hierarchy of control' (*UK Government, 2021b*). Conversely, guidance from the Centers for Disease Control and Prevention (CDC) recommends that HCWs caring for patients with COVID-19 should use an N95 or higher level respirator (*Centers for Disease Control and Prevention, 2019*). WHO guidance suggests that a respirator, such as FFP3, may be used for HCWs in the absence of AGPs if availability or cost is not an issue (*World Health Organization, 2021*).

A recent systematic review undertaken for PHE concluded that: 'patients with SARS-CoV-2 infection who are breathing, talking, or coughing generate both respiratory droplets and aerosols, but FRSM (and where required, eye protection) are considered to provide adequate staff protection' (*Public Health England, 2020*). Nevertheless, FFP3 respirators are more effective in preventing aerosol transmission than FRSMs, and observational data suggest that they may improve protection for HCWs (*Oksanen et al., 2020*). It has therefore been suggested that respirators should be considered as a means of affording the best available protection (*Ha, 2020*), and some organisations have decided to provide FFP3 (or equivalent) respirators to HCWs caring for COVID-19 patients, despite a lack of mandate from local or national guidelines (*Buising et al., 2020*).

Data from the HCW testing programme at Cambridge University Hospitals NHS Foundation Trust (CUHNFT) during the first wave of the UK severe acute respiratory syndrome coronavirus 2 (SARS-CoV-2) pandemic indicated a higher incidence of infection amongst HCWs caring for patients with COVID-19, compared with those who did not (*Rivett et al., 2020*). Subsequent studies have confirmed this observation (*Eyre et al., 2020*; *Cooper et al., 2020*). This disparity persisted at CUHNFT in December 2020, despite control measures consistent with PHE guidance and audits indicating good compliance. The CUHNFT infection control committee therefore implemented a change of RPE for staff on 'red' (COVID-19) wards from FRSMs to FFP3 respirators. In this study, we analyse the incidence of SARS-CoV-2 infection in HCWs before and after this transition.

## Materials and methods

### Study design and participants

CUHNFT is a tertiary hospital in the UK with approximately 1000 beds. During the pandemic, wards were categorised as 'red', 'amber', or 'green'. Patients with confirmed COVID-19 were cared for on red wards, and patients who had negative SARS-CoV-2 tests and no clinical features of COVID-19 on green wards. Patients awaiting test results, who had clinical features of COVID-19 but a negative test result, or who may have been exposed to SARS-CoV-2 were cared for on amber wards.

The CUHNFT electronic rostering system recorded to which ward(s) individual nurses and health-care assistants (HCAs) were allocated. Although this does not encompass 100% of ward staff, the data can be used to indicate relative ward size. An average of 42.5 (range 19–72) nurses/HCAs worked on green wards, and 49.6 (range 37–69) worked on red wards. The mean number of beds per green ward was 24.1 (range 5–33) and red 28.1 (range 26–33). The mean number of nurses and HCAs per bed was 0.41 (range 0.24–0.58) on green wards and 0.31 (range 0.24–0.42) on red wards.

**Table 1.** Weekly numbers of cases amongst HCWs on red and green wards, and cases per HCW day weeks following the change in RPE are highlighted in grey.
Community incidence (total cases per week) is shown for the East of England, UK, with raw data shown in *Figure 1—source data 1*.

| Week | Week start | Red cases | Red HCW days | Red cases per 10³ HCW days | Green cases | Green HCW days | Green cases per 10³ HCW days | Excluded cases | Total | Community |
|------|-----------|-----------|--------------|----------------------------|-------------|----------------|------------------------------|----------------|-------|-----------|
| 1 | 02/11/2020 | 0 | 98 | 0 | 5 | 3255 | 1.54 | 16 | 21 | 7876 |
| 2 | 09/11/2020 | 2 | 98 | 20.41 | 7 | 3241 | 2.16 | 33 | 42 | 9499 |
| 3 | 16/11/2020 | 1 | 198 | 5.05 | 3 | 3141 | 0.96 | 26 | 30 | 7998 |
| 4 | 23/11/2020 | 1 | 238 | 4.20 | 5 | 3101 | 1.61 | 24 | 31 | 7203 |
| 5 | 30/11/2020 | 3 | 238 | 12.61 | 6 | 3101 | 1.93 | 20 | 29 | 9441 |
| 6 | 07/12/2020 | 5 | 238 | 21.01 | 10 | 3101 | 3.22 | 33 | 48 | 16,535 |
| 7 | 14/12/2020 | 1 | 238 | 4.20 | 7 | 3101 | 2.26 | 41 | 49 | 31,219 |
| 8 | 21/12/2020 | 3 | 238 | 12.61 | 10 | 3101 | 3.22 | 56 | 69 | 37,259 |
| 9 | 28/12/2020 | 2 | 357 | 5.60 | 20 | 2982 | 6.71 | 58 | 80 | 50,110 |
| 10 | 04/01/2021 | 4 | 505 | 7.92 | 34 | 2834 | 12.00 | 70 | 108 | 41,663 |
| 11 | 11/01/2021 | 5 | 848 | 5.90 | 33 | 2491 | 13.25 | 63 | 102 | 31,341 |

HCW = healthcare worker. RPE = respiratory protective equipment.

A change to RPE for staff on red wards from FRSMs to FFP3 respirators was announced on 22/12/20. FFP3 respirators were assigned to staff following fit testing. HCWs on green wards continued to wear FRSMs. HCWs on all wards also wore eye protection. The following types of FFP3 respirator were used during the study period: 3 M 9330+, 3 M 1863, Easimask FSM18, and Mexin MX2016v. HCWs who did not pass fit testing with the masks available used either a JSP half mask respirator or a powered air purifying respirator (Tornado or Easiair).

A comprehensive PCR-based HCW screening programme is established at CUHNFT, with symptomatic testing offered as required and asymptomatic testing offered to all HCWs weekly (*Rivett et al., 2020*; *Jones et al., 2020*). From 22/12/20, twice-weekly swabbing was offered on red wards and on wards where the most vulnerable patients were cared for (e.g. transplant and oncology patients). Cases were identified from a database of all positive results, which additionally encompasses positive results from community testing. This recorded the date of swab, onset of symptoms (if present) and in which clinical area the HCW worked.

The start of the study period was taken to be 02/11/20, coinciding with an increase in community incidence of SARS-CoV-2 infection and formal implementation of weekly asymptomatic screening for all staff members. By default new infections on or prior to 27/12/20 were attributed to exposure before the change in RPE. Infections detected later than this date were attributed to exposure after the change in RPE. This timing was chosen to reflect the median incubation period of SARS-CoV-2 (5.1 days), with 27/12/20 falling 5 days (inclusive) after the change in RPE (*Lauer et al., 2020*; *McAloon et al., 2020*). Since staff testing was not conducted at weekends, eight complete weeks were assessed in total prior to the change in RPE (*Table 1*).

A programme of SARS-CoV-2 vaccination using the BNT162b2 COVID-19 vaccine commenced at CUHNFT on 08/12/20 (*Jones et al., 2021*). In line with UK national guidance, the programme initially prioritised local residents over the age of 80. However, some HCWs who had been identified as at high risk from SARS-CoV-2 infection were also vaccinated, and were additionally prevented from working on red wards. From 08/01/21, the programme switched to vaccinating HCWs, with initial priority being given to staff on red wards. To avoid the potential for confounding, the final week of the

study period commenced on 11/01/21, since minimal effect is expected in the first 7 days after the first dose of vaccine (**Polack et al., 2020**).

Because of the rising number of admissions to CUHNFT with COVID-19, the number of red wards was increased from one at the beginning of November 2020 to seven by the week starting 11/01/21. Six wards therefore changed from green to red during the period of data collection. Of 609 positive results over the entire study period, 169 (27.8%) were included in this study. Exclusions encompassed HCWs who were not ward based or worked between different wards with different red/amber/green status (269/609, 44.2 % of positive results), HCW working on amber wards (9/609, 1.5%), non-clinical staff (141/609, 23.1%), and staff working in critical care areas (21/609, 3.5%), where different RPE was used throughout (**Table 1**).

If a staff member tested positive within 5 days of their ward changing colour, their case was classified according to the red/green status of their ward 5 days before their positive test (to allow for the incubation period, as above). The effects of changing the interval from 5 days to between 3 and 7 days explored.

## General statistical analysis

The number of 'HCW days' for each week of the study was calculated for each category of ward. Rostering information was used to identify the number of nurses and HCAs regularly assigned to each ward on each of the 7 days of the week. Data describing the number of other staff on each ward was not available, but was assumed to be proportional to the number of rostered HCWs, calculations being performed in terms of nurse and HCA numbers.

Where $w_{X,d}$ denotes the number of HCWs on wards of type $X$ on day $d$, the weekly numbers of ward days for week $i$, denoted $W_{X,i}$, were calculated as the sums of these values across that week.

$$W_{X,i} = \sum_{d \in i} w_{X,d}$$

Details of community incidence were calculated from publicly available data describing the East of England region of the UK (**Wellcome Sanger Institute, 2021**; https://coronavirus.data.gov.uk/details/cases, data downloaded on 12/06/21), and were calculated as the sum of the number of cases reported in each week of the study. Raw data are shown in **Figure 1—source data 1**. Correlations between cases per ward day and community incidence were calculated using the Wolfram Mathematica software package, version 12.3.1.0.

## Mathematical modelling

In order to quantify the effect of the change in RPE upon cases in red wards, a mathematical model was developed, considering the numbers of cases observed amongst HCWs as arising from a combination of ward-specific infection risks, which relate directly to working on a red or green ward, and non-ward-specific risks, which include infections arising from the community. We first wrote expressions for the infection risk facing workers in different types of wards on week $i$. For HCWs on green wards we write

$$\lambda_i^G = \left(kC_{i-1} + g\right) W_{G,i}$$

Where critical care wards were included in the model we write, similarly:

$$\lambda_i^C = \left(kC_{i-1} + c\right) W_{C,i}$$

Cases on red wards were split according to whether they arose prior to the introduction of FFP3 masks ($R_1$) or after that point ($R_2$), giving:

$$\lambda_i^{R_1} = \left(kC_{i-1} + r_1\right) W_{R_1,i}$$
$$\lambda_i^{R_2} = \left(kC_{i-1} + r_1\right) W_{R_2,i}$$

Here, the term $k$ is a constant, whilst the value $C_{i-1}$ describes the number of observed cases in the local community in the previous week. Our use of community data from the previous week reflects a generation time for SARS-CoV-2 of approximately 7 days (**Volz et al., 2021**); we assumed that HCWs diagnosed with COVID-19 infection during this study would have been infected by individuals who

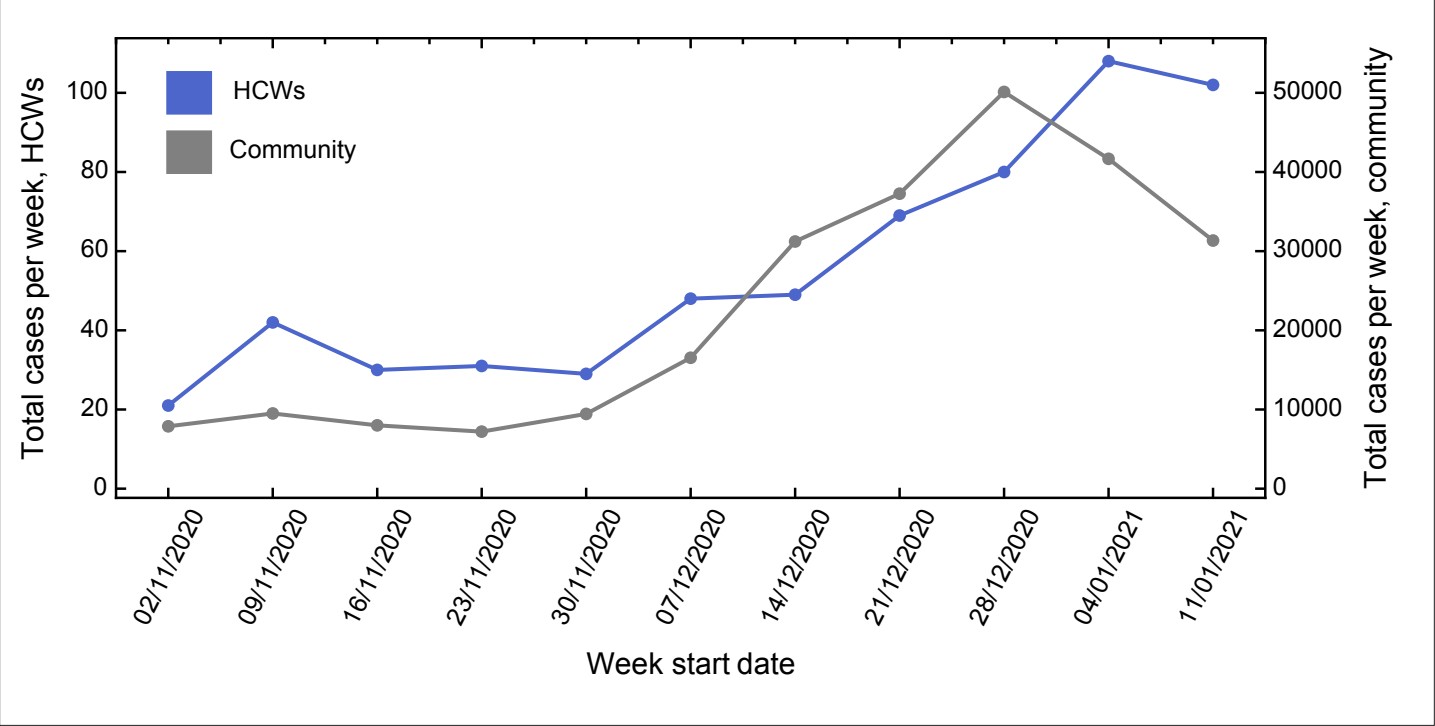

**Figure 1.** Comparison between total number of cases amongst healthcare workers (HCWs) and community incidence of severe acute respiratory syndrome coronavirus 2 (SARS-CoV-2). Comparison between total number of cases amongst HCWs and community incidence of SARS-CoV-2. Community incidence is shown for the East of England, UK, derived from https://coronavirus.data.gov.uk/details/cases, with raw data shown in *Figure 1—source data 1*.

The online version of this article includes the following source data and figure supplement(s) for figure 1:

**Source data 1.** Raw case numbers for the East of England region during the period of study.

**Figure supplement 1.** Proportion of cases ascertained by symptomatic testing and asymptomatic screening on green and red wards.

**Figure supplement 1—source data 1.** Proportion of cases ascertained by symptomatic and asymptomatic screening on green and red wards.

**Figure supplement 2.** Relationship between number of healthcare worker (HCW) days per week worked on red wards and community incidence.

were diagnosed in the previous week. The model parameters $g$, $c$, $r_1$, and $r_2$ describe ward-specific infection risks. FFP3 masks were used from the 23rd December onwards.

Model parameters were optimised using a likelihood framework, identifying the maximum value of the term; here, the number of cases on each type of ward each week, denoted $X_i$, was represented as emissions from a Poisson distribution with parameter equal to the total risk of infection.

$$L = \sum_i \left[ \sum_X log \frac{\lambda_i^{X_i}}{X_i!} \right]$$

where the sum inside the brackets was calculated over all ward types $X$.

Confidence intervals for each parameter were obtained using this likelihood function. Constrained likelihood optimisations were performed in which the likelihood was optimised subject to a fixed value of the parameter in question. Confidence intervals were defined as the region of parameter space in which the likelihood $L$ was within 2 units of the maximum. Similarly, constrained optimisation was used to identify confidence intervals for parameter ratios such as $r_2/r_1$.

## Results

The total number of cases of SARS-CoV-2 infection amongst HCWs at CUHNFT increased throughout the study period, in keeping with the rising incidence of SARS-CoV-2 in the community (*Figure 1* and *Figure 1—source data 1*). Similar proportions of cases were ascertained by symptomatic testing and asymptomatic screening on both green and red wards, suggesting similar testing-seeking behaviour

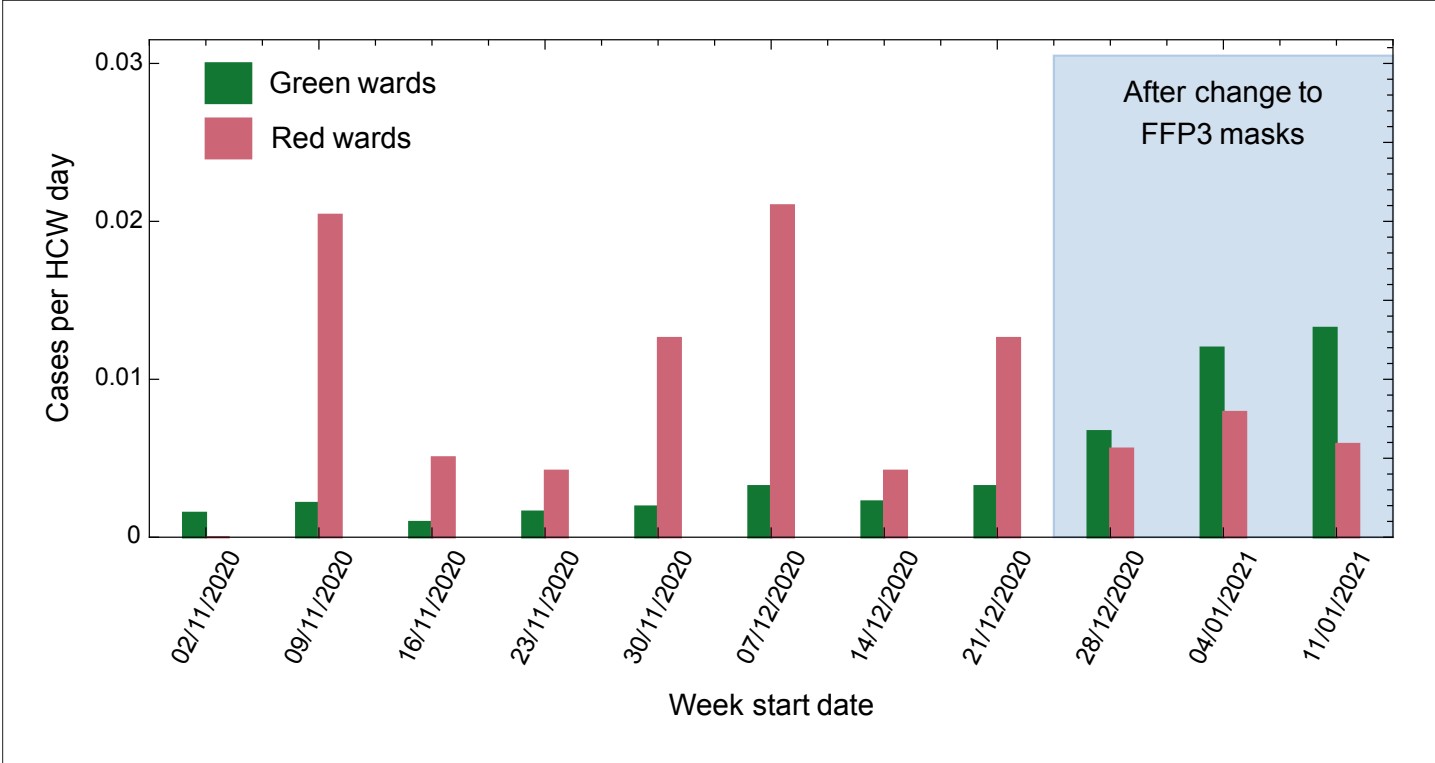

**Figure 2.** Weekly cases per healthcare worker (HCW) day amongst HCWs on red and green wards prior to and after the change in respiratory protective equipment (RPE).

The online version of this article includes the following figure supplement(s) for figure 2:

**Figure supplement 1.** Relationships between cases per ward day and community incidence.

between staff groups (*Figure 1—figure supplement 1*). 12.1 % of cases on green wards were amongst allied health professionals, such as physiotherapists and occupational therapists. As expected, there was a significant correlation between community cases and days worked by HCWs on red wards (p < 0.002, Pearson correlation test), reflecting increased hospital admissions (*Figure 1—figure supplement 2*).

Prior to the change in RPE, cases per HCW day were higher on red compared with green wards in seven out of 8 weeks analysed (p = 0.016, Wilcoxon signed-rank test, *Figure 2* and *Table 1*). Following the change in RPE, the incidence of infection on red and green wards was similar, and not statistically different (p = 0.5, Wilcoxon signed-rank test, *Figure 2* and *Table 1*). Strikingly, there was a strong positive correlation between the incidence of SARS-CoV-2 in the community and the number of cases per HCW day on green ($R^2 = 0.80$) but not red ($R^2 = 0.03$) wards (*Figure 2—figure supplement 1*). Taken together, these results suggest that most cases amongst HCWs on green wards were caused by community-acquired infection, whereas cases amongst HCWs on red wards were driven by direct, ward-based infection from patients with COVID-19.

To further quantify the risk of infection for HCWs working on red and green wards, we generated a simple mathematical model. According to this model, the total risk of infection is divided into a risk from community-based exposure, and a risk from direct, ward-based exposure to patients (ward-specific risk). The risk from direct exposure on red wards was allowed to vary upon the introduction of FFP3 respirators, and was fitted to a maximum likelihood model. Inferred parameters and their confidence intervals are shown in *Table 2*. Our model produced a qualitatively close fit to the observed numbers of cases (*Figure 3A, B*).

The inferred risk of direct infection from working on a green ward was low throughout the study period, and consistently lower than the risk of community-based exposure, which increased in proportion to rising levels of community incidence (*Figure 3C*). By contrast, the risk of direct infection from working on a red ward before the change in RPE was considerably higher than the risk

**Table 2.** Statistics and parameter ratios inferred from the model.

| Statistic | Model parameter | Maximum likelihood estimate | Confidence interval |
|---|---|---|---|
| Force of community-based infection per community case | $k$ | $1.95 \times 10^{-7}$ | $[1.49 \times 10^{-7}, 2.39 \times 10^{-7}]$ |
| Force of direct infection per HCW day (green ward) | $g$ | $2.53 \times 10^{-4}$ | $[0, 1.10 \times 10^{-3}]$ |
| Force of direct infection per HCW day (red ward, pre-FFP3) | $r_1$ | $7.97 \times 10^{-3}$ | $[3.65 \times 10^{-3}, 1.40 \times 10^{-2}]$ |
| Force of direct infection per ward day (red ward, post-FFP3) | $r_2$ | $6.84 \times 10^{-10}$ | $[0, 3.38 \times 10^{-3}]$ |
| Relative direct risk on red wards post- versus pre-FFP3 | $r_2/r_1$ | 0.00 | $[0, 0.478]$ |
| Relative direct risk on red ward versus green ward pre-FFP3 | $r_1/g$ | 31.47 | $[5.92, \infty)$ |
| Relative direct risk on red ward versus green ward post-FFP3 | $r_2/g$ | 0.00 | $[0, \infty)$ |

FFP3 = filtering face piece 3. HCW = healthcare worker.

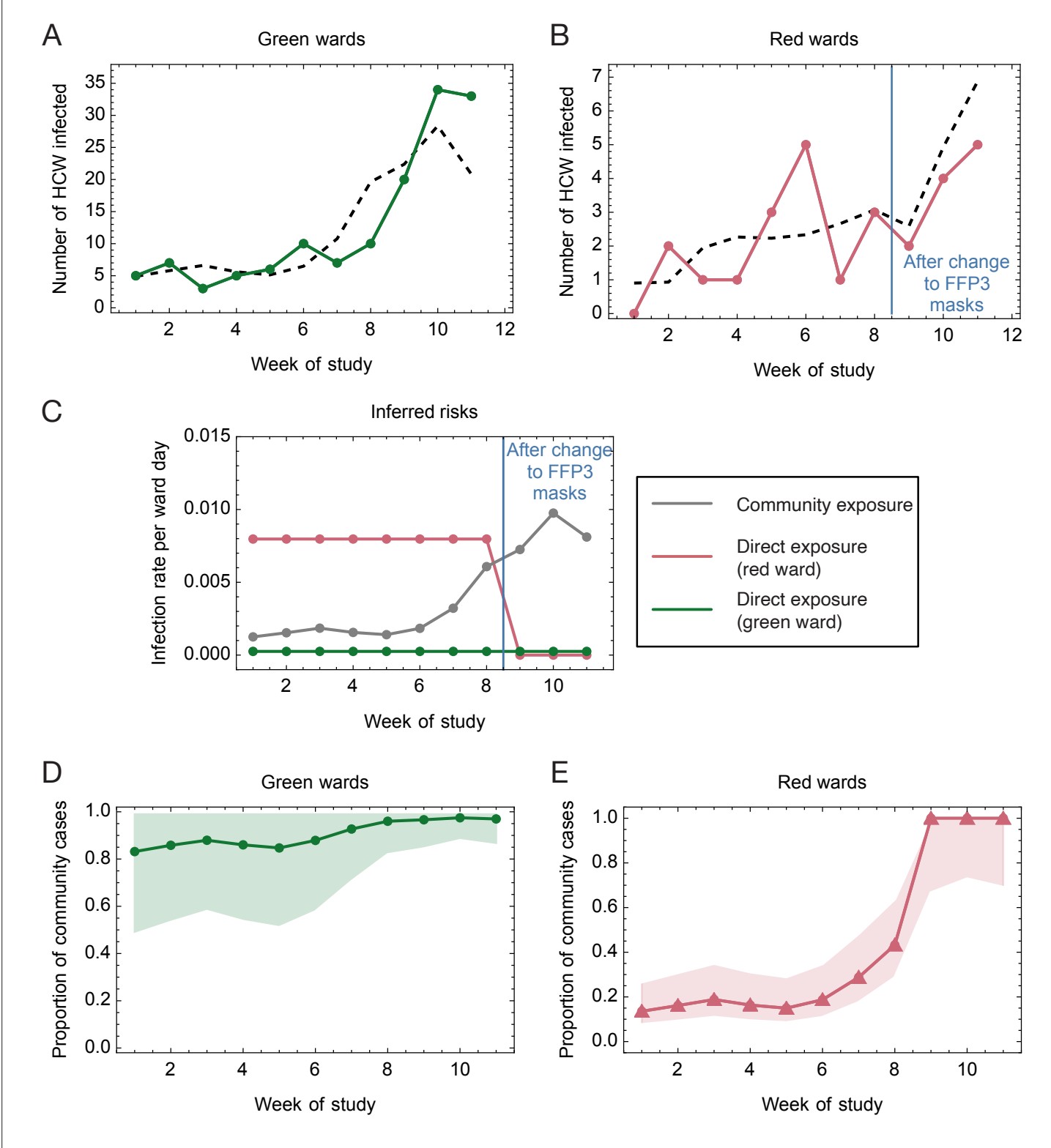

**Figure 3.** Mathematical modelling of the risks of infection for healthcare workers (HCWs) on red and green wards. (**A, B**) Comparison of modelled and actual cases. The model (black dashed line) aimed to reproduce the risks of infection amongst HCWs per ward day (**A**) on green wards (green solid line) and (**B**) on red wards (red solid line). (**C**) Risks inferred from the model. HCWs were vulnerable to coronavirus disease 2019 (COVID-19) infection from exposure to individuals in the community, with this risk increasing with community incidence (grey line). HCWs working on green wards faced a consistent, low risk of infection from direct, ward-based exposure (green line). HCWs working on red wards initially faced a much higher risk of infection

*Figure 3 continued on next page*

*Figure 3 continued*

from direct, ward-based exposure, falling to a value close to that on green wards upon the introduction of filtering face piece 3 (FFP3) respirators. In this figure, risks are expressed per ward day; a risk of 0.01 indicates that a particular source of risk would be expected to cause one HCW to develop an infection every 100 days that the ward was in operation. (**D, E**) Proportion of community-acquired cases. Proportion of infections on (**D**) green and (**E**) red wards inferred to have arisen via exposure to individuals in the community (green line, green wards; red line, red wards; confidence intervals shaded).

The online version of this article includes the following source data and figure supplement(s) for figure 3:

**Source data 1.** Mathematical modelling of the risks of infection for healthcare workers (HCWs) on red and green wards.

**Figure supplement 1.** Effect of changing the attribution of positive cases to wards in which a contemporaneous designation change occurred (e.g. from green to red).

**Figure supplement 1—source data 1.** Effect of changing the attribution of positive cases to wards in which a contemporaneous designation change occurred.

**Figure supplement 2.** Comparison of modelled and actual cases when critical care wards were included in the dataset.

**Figure supplement 2—source data 1.** Comparison of modelled and actual cases when critical care wards were included in the dataset.

of community-based exposure, and approximately 31-fold greater than the corresponding risk from working on a green ward (confidence interval [5.93, ∞]). Thus, whilst a high proportion of cases on green wards were likely caused by infection in the community, cases on red wards at the beginning of the study period were attributed mainly to direct, ward-based exposure (**Figure 3D, E**). Critically, our model further suggests that the introduction of FFP3 respirators led to a reduction of between 52% and 100% (maximum likelihood 100%) in the risk of direct, ward-based COVID-19 infection (**Table 2**, $r_2/r_1$).

Where ward designations changed (e.g. from green to red), cases were by default attributed to the type of ward on which each positive-testing HCW worked 5 days prior to reporting symptoms (if symptomatic) or testing positive (if asymptomatic). Altering this cutoff did not alter the maximum likelihood inference for the effect of FFP3 respirators ($r_2/r_1$, 100%), although the lower bound of the effect size varied between 30% and 72% for cutoffs between 3 and 7 days (**Figure 3—figure supplement 1**). Data collected from critical care wards, where enhanced PPE was used throughout the period of the study, showed a consistently low rate of HCW infection. Again, incorporating these data into the model did not materially affect the outcome, with the introduction of FFP3 respirators associated with a reduction of between 26% and 100% (most likely 94%) in the risk of direct, ward-based COVID-19 infection at the default cutoff (**Figure 3—figure supplement 2**).

## Discussion

HCWs may be exposed to SARS-CoV-2 from contacts in the community, from contacts with other HCWs, and from contacts with patients. In this study, we developed a mathematical model to evaluate the relative magnitudes of these risks, based on data collected during the second wave of the SARS-CoV-2 pandemic in the UK (November 2020–January 2021).

Whilst using FRSMs, the majority of infections amongst HCWs working on red wards could be attributed to direct exposure to patients with COVID-19. In contrast, as community incidence rose, the majority of infections amongst HCWs working on green wards were attributed by our model to community-based effects. After the change in RPE, cases attributed to ward-based exposure fell significantly, with FFP3 respirators providing an inferred 52–100% (most likely 100%) reduction in the risk of ward-based infection from patients with COVID-19.

In keeping with previous observations (**Rivett et al., 2020**; **Eyre et al., 2020**; **Cooper et al., 2020**), our findings therefore suggest that the use of FRSMs as RPE was insufficient to protect HCWs against infection from patients with COVID-19. Conversely, excess infections amongst HCWs caring for patients with COVID-19 may be prevented by the use of FFP3 respirators, in combination with other PPE and infection control measures.

During the study period, the incidence of SARS-CoV-2 in England increased (**Office of National Statistics, 2021**), with spread of the more transmissible B.1.1.7 (alpha) variant (**Davies et al., 2021**). By the ninth week of the study, 79 % of cases in Cambridgeshire were caused by this variant (**Wellcome Sanger Institute, 2021**). Our observations on the use of FFP3 respirators (weeks 9–11) were therefore made at a time when the B.1.1.7 variant predominated, suggesting that they are robust to

any associated increase in SARS-CoV-2 transmissibility in a hospital setting attributable to this variant. Whilst likely also to be applicable to the B.1.617.2 (delta) variant, this was not formally evaluated in our study.

Potential confounders of our observations, should they have differed systematically between HCWs on red and green wards and/or have changed over the course of the study, include:

1. Rates of natural immunity amongst HCWs on red and green wards; however, the frequency of prior SARS-CoV-2 infections was low within CUHNFT. Overall seropositvity revealed by testing in July and August 2020 was 7.2 % (9.47 % amongst staff from red wards versus 6.16 % amongst all other staff) (*Cooper et al., 2020*).
2. Rates of vaccination of HCWs on red and green wards; however, the proportion of high-risk HCWs at CUHNFT offered vaccination prior to 08/01/21 was very low, and the study period was ended on 17/01/21 (before any substantial impact of vaccination was expected).
3. Frequency of asymptomatic screening of HCWs on red and green wards; however, the proportions of cases ascertained by symptomatic testing versus asymptomatic screening were similar in both settings. In addition, whilst twice-weekly testing was available for red ward staff from week 8 of the study, this would have tended to increase (rather than decrease) the ascertainment of HCW cases on red wards after the change in RPE in week 9.
4. Compliance with infection control measures by HCWs on red and green wards. It is possible that some of the effect of the change in RPE may have been mediated indirectly, by triggering changes in other behaviours; however, this would still be a positive outcome.
5. Exclusion of infections amongst HCWs who worked on wards from multiple categories (such as, both green and red wards); however, this would have tended to minimise any difference in ward-specific risk of infection.
6. Differences in patterns of HCW behaviour on red and green wards, including mixing between HCWs from different areas. For example, staff working on green wards may have been more likely to leave the ward for lunch than staff working on red wards. Whilst such differences could in theory have contributed to the greater risk of HCW infection on red wards, they are unlikely to have changed systematically with the change in RPE. In addition, if mixing between HCWs from different areas led to an increased rate of infection, it would have tended to minimise any difference in ward-specific risk of infection.

This observational study includes a small number of cases in a single Trust, and there may be alternative explanations for the different patterns of infection observed before and after the change in RPE. Our maximum likelihood inference that FFP3 masks (in combination with other PPE and infection control measures) provide 100 % protection against ward-based infection should therefore be treated with caution; the large confidence intervals calculated for parameters in our model reflect the limited amount of data available. Nonetheless, our results highlight an urgent need for further studies evaluating the appropriate level of RPE for HCWs caring for patients with COVID-19, as well as other respiratory viruses. In accordance with the precautionary principle, we propose a revision of RPE recommendations until more definitive information is available.

## Acknowledgements

This research was funded in part by a Wellcome Trust Senior Clinical Research Fellowship [108070/Z/15/Z] and grants from Addenbrooke's Charitable Trust and the NIHR Cambridge Biomedical Research Centre to MPW. NJM was funded by an MRC Clinician Scientist Fellowship [MR/P008801/1] and NHSBT workpackage [WPA15-02]. CJRI was supported by UKRI through the JUNIPER modelling consortium [MR/V038613/1] and by the Medical Research Council [MC_UU_00002/11, MC_UU_12014]. For the purpose of open access, the authors have applied a CC-BY public copyright licence to any author accepted manuscript version arising from this submission. We would like to thank everyone involved in the development and operation of the SARS-CoV-2 testing programme at CUH and the members of staff who have participated. We would also like to thank the Infection Control and Fit Testing teams.

## Additional information

### Funding

| Funder | Grant reference number | Author |
|---|---|---|
| Wellcome Trust | 108070/Z/15/Z | Michael P Weekes |
| Addenbrooke's Charitable Trust, Cambridge University Hospitals | | Michael P Weekes |
| NIHR Cambridge Biomedical Research Centre | | Michael P Weekes |
| Medical Research Council | MR/P008801/1 | Nicholas J Matheson |
| NHS Blood and Transfusion | WPA15-02 | Nicholas J Matheson |
| UK Research and Innovation | MR/V038613/1 | Christopher JR Illingworth |
| Medical Research Council | MC_UU_00002/11 | Christopher J R Illingworth |
| Medical Research Council | MC_UU_12014 | Christopher J R Illingworth |

The funders had no role in study design, data collection and interpretation, or the decision to submit the work for publication.

### Author contributions

Mark Ferris, Conceptualization, Data curation, Formal analysis, Investigation, Methodology, Project administration, Supervision, Writing – original draft, Writing – review and editing; Rebecca Ferris, Data curation, Formal analysis, Investigation, Validation, Visualization, Writing – original draft, Writing – review and editing; Chris Workman, Data curation, Formal analysis, Visualization, Writing – review and editing; Eoin O'Connor, Data curation, Formal analysis, Writing – review and editing; David A Enoch, Conceptualization, Investigation, Methodology, Writing – review and editing; Emma Gold-esgeyme, Investigation, Writing – original draft, Writing – review and editing; Natalie Quinnell, Data curation, Formal analysis, Project administration, Writing – review and editing; Parth Patel, Data curation, Writing – original draft, Writing – review and editing; Jo Wright, Data curation, Formal analysis, Investigation, Project administration, Writing – review and editing; Geraldine Martell, Investigation, Writing – review and editing; Christine Moody, Data curation, Investigation, Writing – review and editing; Ashley Shaw, Investigation, Methodology, Writing – original draft, Writing – review and editing; Christopher JR Illingworth, Conceptualization, Data curation, Formal analysis, Investigation, Software, Visualization, Writing – original draft, Writing – review and editing; Nicholas J Matheson, Michael P Weekes, Conceptualization, Data curation, Formal analysis, Funding acquisition, Investigation, Methodology, Supervision, Visualization, Writing – original draft, Writing – review and editing

### Author ORCIDs

Mark Ferris ⓘ http://orcid.org/0000-0001-5040-4263
Eoin O'Connor ⓘ http://orcid.org/0000-0002-6846-6881
Christopher JR Illingworth ⓘ http://orcid.org/0000-0002-0030-2784
Nicholas J Matheson ⓘ http://orcid.org/0000-0002-3318-1851
Michael P Weekes ⓘ http://orcid.org/0000-0003-3196-5545

### Ethics

Human subjects: This study was conducted as a service evaluation of the CUHNFT staff testing services and PPE policy (CUHNFT clinical project ID3738). As a study of healthcare-associated infections, this investigation is exempt from requiring ethical approval under Section 251 of the NHS Act 2006 (see also the NHS Health Research Authority algorithm, available at http://www.hra-decision-tools.org.uk/research/, which concludes that no formal ethical approval is required).

### Decision letter and Author response

Decision letter https://doi.org/10.7554/eLife.71131.sa1
Author response https://doi.org/10.7554/eLife.71131.sa2

## Additional files

### Supplementary files
- Transparent reporting form
- Supplementary file 1. Additional data tables.

### Data availability
All data generated or analysed during this study are included in the manuscript and supporting files. Source data files have been provided for Figures 1 and 3, and their supplements. Figure 2 source data is included in Table 1 in the main text.

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
