## [Editor Report]

Respiratory protective equipment that is recommended in the UK for health-care workers caring for COVID-19 patients comprises a fluid resistant surgical mask (FRSM), and in case of procedures that generate aerosols FFP3 respirators are to be used. In this study, health-care workers using FRSMs, while working on COVID-19 wards faced an approximately 31-fold increased risk of ward-based SARS CoV-2 infection. After changing to FFP3 respirators, this risk was significantly reduced. Thus, FFP3 respirators seem to provide more protection than FRSMs for health-care workers caring for patients with COVID-19.

---

## [Decision Letter]

**Decision letter after peer review:**

Thank you for submitting your article "FFP3 respirators protect healthcare workers against infection with SARS-CoV-2" for consideration by *eLife*. Your article has been reviewed by 2 peer reviewers, and the evaluation has been overseen by a Reviewing Editor and Jos van der Meer as the Senior Editor. The following individuals involved in review of your submission have agreed to reveal their identity: Sarah Logan (Reviewer #1); Stephanie Evans (Reviewer #2).

Essential revisions:

1. The authors need to mention more about the differences between red and green wards: In terms of:

Number of patients on the wards (bays and side rooms)

Mean number of pts per nurse

Demographic of the HCW surveyed on each ward- equal numbers of Nurses/ AHP's/ doctors

Vaccine uptake as per below.

Staff rest areas.

2. The authors also need to describe more about the testing seeking behaviour of the two staff groups; access to testing, more frequently accessed by a group of HCW who were aware of the range of symptoms of covid. This could leave to an ascertainment bias in the red ward prior to the change in mask wearing.

3. Vaccine roll out:

Most staff working on COVID wards in December 2020 were the first to access the vaccine, and the roll out in most trusts in the UK was not so coordinated. Thus, we imagine this data will be difficult to get, but without it, more needs to be made of that impact.

4. Given the sophisticated mathematical modelling and statistical analysis, please address the danger of trying to overanalyse the intervention with relatively small numbers.

5. With this in mind it would be nice to see data from critical care ward staff, if they were using FFP3 throughout even though the numbers are small.

6. We would recommend to pay some attention in the discussion to the comments from Andrew Goddard from the RCP bulletin.

7. Although the manuscript states that a similar number of HCWs worked on each ward, it would be helpful to see the results displayed as a proportion of staff on red/green wards that became infected, or to model as individual exposure hours rather than ward status but I understand this might be difficult with the small sample size.

8. It would be helpful for the number of red wards (or total exposure time) and C_i+1 to be plotted to give a visual representation of the correlation.

9. As ward colour in the model is determined exclusively by the presence of infected patients and there is a possibility that one infected HCW on a green ward could lead to an outbreak within the HCW population. While this would not affect the result that changing the type of RPE used reduced the infections per ward day of HCWs on red wards, ignoring the potential for HCWs to seed outbreaks between ward staff could result in a higher proportion of cases being attributed to the community than actually occurred, and I would consider adding a third component of 'positive result in ward staff' to the model.

10. A sensitivity analysis shifting the 5-day cut off for before/after change in RPE and ward classification would also be a nice addition.

11. As staff working across wards are excluded, it is not clear if the full numbers of staff present on the red/green wards are included in estimations of both the number of cases and number of infectious sources (do the staff, and the non-ward based staff, interact similarly with patients and staff from both red and green wards?).

12. It would be interesting to see the results from the critical care wards that are currently excluded. If these wards used FFP3 respirators throughout, (noting the small numbers) it would be very helpful to see the data for this period in these wards as a (albeit limited) control group.*Reviewer #1:*

Understanding how best to protect healthcare workers from infection at work with SARS- CoV2 is absolutely vital. Much of the interventions to date have been done in an era where the evidence was emerging at the same time as interventions instigated. With that in mind the approach by Cambridge University trust to go above PHE guidance and use FFP3 masks on COVID wards and then for the authors to try to evaluate this impact is absolutely the right approach. There is a clear drop off in infections in healthcare workers on COVID wards in December 2020 after this change was made. There may be several reasons for this not just the prevention of inhalation of aerosols.

The action of donning and doffing an FFP3 mask produces a change in behaviour in the HCW. The need to change one's mask after leaving a clinical area, the discomfort of an FFP3 necessitating shorter wear times than alternative Fluid resistant surgical masks, the prompt to the HCW when changing the mask of hand hygeine may all have contributed to the drop in rate.

The drop-in rate correlates also with vaccine role out. Whilst the authors describe this as not having a major impact in the absence of data on the numbers and date of vaccination of staff on the red and green wards it is impossible to draw this conclusion.

To be fair, the authors themselves tried not to sensationalise their data but I suspect that many infection control teams will have noted the study and (reasonably) will have pressure put on them to act if they have not done so already. Understanding the risks in double-vaccinated staff, ongoing supply chain issues and the fact that at least 20% of FFP3 mask wearers fail a fit test may mean that applying a simple policy to change all to FFP3 will not be without practical challenges.*Reviewer #2:*

In this paper Ferris et al., attempt to use observational data and mathematical modelling to compare the effect of filtering face piece 3 (FFP3) respirators over fluid resistant surgical masks (FRSM) for reducing infection rates in healthcare workers (HCWs) caring for patients with COVID-19 (working on 'red' wards).

The study uses the median incubation period of 5 days to classify new cases as occurring before/after the change in respiratory protective equipment (RPE), and to categorise wards as red and green. This goes some way towards addressing potential confounding that could be introduced by misclassifying either RPE or wards, although this cannot be completely ruled out. Similarly the study is over a sufficient time period that it is reasonable to assume that the effect of vaccines in the HCW population is negligible. Overall the modelling approach, while simple (with only two components of risk: community and ward), is sensible. The Discussion section of this work is fair and balanced, and the concluding statement that this study is not sufficient to provide conclusive evidence but highlights a need for further study into the appropriate RPE for HCWs caring for patients with COVID-19 is faultless.

As explained by the authors the main limitations of the study hinge upon the small numbers. While the model itself is broadly sensible, the fitting of this to the very limited data on HCW infection is problematic. Slightly different assumptions (e.g. changing the criteria for classifying wards from the median) may provide an alternative (but equally good) fit, but give different conclusions.

A further limitation of the model used here is the association between Ci-1 and determination of ward 'colour': Ci-1 will determine how many green and red wards there are. Likewise, community prevalence will determine the number of COVID +ve admissions which in turn may impact onward transmission within the hospital, therefore Ci-1 is associated too with ward risk.

---

## [Author Response]

Essential revisions:1. The authors need to mention more about the differences between red and green wards: In terms of:

Many thanks for this point. We have added additional information under the subheadings below:

Number of patients on the wards (bays and side rooms)

We have added these details to the Methods section, detailing the mean and ranges for numbers of beds on red and green wards:

“The mean number of beds per green ward was 24.1 (range 5 to 33) and red 28.1 (range 26 to 33).”

Mean number of pts per nurse

This data has been added to the Methods section:

“The mean number of nurses and HCAs per bed were 0.41 (range 0.24 to 0.58) on green wards and 0.31 (range 0.24 to 0.42) on red wards.”

Demographic of the HCW surveyed on each ward- equal numbers of Nurses/ AHP's/ doctors

The table details the job roles of the positive-testing HCWs included in the study. It has been included in the paper as Supplementary file 1.

This data suggests that a higher proportion of cases (85.2% versus 77.9%) fell amongst nurses and healthcare assistants (HCAs) for red compared to green wards, and that there were no allied health professional (AHP) cases on red wards but 17 (12.1%) cases on green wards. This difference is understandable, because AHPs tend to work on multiple wards, there were many more green than red wards, and HCWs who worked on both red and green wards were excluded from the study.

Vaccine uptake as per below.

We have clarified in the paper that the HCW vaccination programme did not commence until 08/01/21. We selected 11/01/21 to be the start of the final week of data analysis to mitigate against vaccination confounding results. The proportion of high-risk HCWs at CUHNFT offered vaccination prior to 08/01/21 was very low.

Clarification of this is included in two sections of the paper:

Discussion

“(b) Rates of vaccination of HCWs on red and green wards; however, the proportion of high-risk HCWs at CUHNFT offered vaccination prior to 08/01/21 was very low, and the study period was ended on 17/01/21 (before any substantial impact of vaccination was expected).”

Methods

“A programme of SARS-CoV-2 vaccination using the BNT162b2 COVID-19 vaccine commenced at CUHNFT on 08/12/20 [15]. In line with UK national guidance, the programme initially prioritised local residents over the age of 80. However, some HCWs who had been identified as at high risk from SARS-CoV-2 infection were also vaccinated, and were additionally prevented from working on red wards. From 08/01/21 the programme switched to vaccinating HCWs, with initial priority being given to staff on red wards. To avoid the potential for confounding, the final week of the study period commenced on 11/01/21, since minimal effect is expected in the first seven days after the first dose of vaccine [16].”

Staff rest areas.

Similar facilities were available on both red and green wards, and there were no changes in rest areas prior to and after the change in PPE. One caveat is that green ward staff were more likely to leave the ward for lunch than red ward staff, however if an increased rate of infections amongst green ward staff resulted, this would tend to underestimate the difference between both ward types. We have added this to the ‘limitations’ section of our discussion as follows:

“(f) Differences in patterns of HCW behaviour on red and green wards, including mixing between HCWs from different areas. For example, staff working on green wards may have been more likely to leave the ward for lunch than staff working on red wards. Whilst such differences could in theory have contributed to the greater risk of HCW infection on red wards, they are unlikely to have changed systematically with the change in RPE. In addition, if mixing between HCWs from different areas led to an increased rate of infection, it would have tended to minimise any difference in ward-specific risk of infection.”

2. The authors also need to describe more about the testing seeking behaviour of the two staff groups; access to testing, more frequently accessed by a group of HCW who were aware of the range of symptoms of covid. This could leave to an ascertainment bias in the red ward prior to the change in mask wearing.

Many thanks for this point. Testing seeking behaviour of the two staff groups was very similar. This was evidenced by the similar proportions of cases ascertained by symptomatic testing and asymptomatic screening on both green and red wards (Figure 1—figure supplement 1). We have already highlighted this point in the Results section, however have now clarified this point further, modifying the second sentence of the results to read:

“Similar proportions of cases were ascertained by symptomatic testing and asymptomatic screening on both green and red wards suggesting similar testing-seeking behaviour between staff groups (Figure 1—figure supplement 1).”

Furthermore, access to testing was identical in the initial part of the study, with symptomatic testing offered as required and asymptomatic testing offered to all HCWs weekly. From 22/12/20, twice-weekly swabbing was offered on red wards and on wards where the most vulnerable patients were cared for. We have already stated these points in the Methods section of the paper, however have now added as a limitation:

“Potential confounders of our observations, should they have differed systematically between HCWs on red and green wards and/or have changed over the course of the study, include: …… (c) Frequency of asymptomatic screening of HCWs on red and green wards; however, the proportions of cases ascertained by symptomatic testing versus asymptomatic screening were similar in both settings. In addition, whilst twice-weekly testing was available for red ward staff from week 8 of the study, this would have tended to increase (rather than decrease) the ascertainment of HCW cases on red wards after the change in RPE in week 9.”

3. Vaccine roll out:Most staff working on COVID wards in December 2020 were the first to access the vaccine, and the roll out in most trusts in the UK was not so coordinated. Thus, we imagine this data will be difficult to get, but without it, more needs to be made of that impact.

Many thanks for requesting this clarification. We have already detailed how vaccination was prioritised in the Methods section:

“A programme of SARS-CoV-2 vaccination using the BNT162b2 COVID-19 vaccine commenced at CUHNFT on 08/12/20 [15]. In line with UK national guidance, the programme initially prioritised local residents over the age of 80. However, some HCWs who had been identified as at high risk from SARS-CoV-2 infection were also vaccinated, and were additionally prevented from working on red wards. From 08/01/21 the programme switched to vaccinating HCWs, with initial priority being given to staff on red wards. To avoid the potential for confounding, the end of the study period was therefore taken to be 17/01/21, since minimal effect is expected in the first seven days after the first dose of vaccine [16].”

To clarify this point, we have added details in the discussion about potential confounders of our observations:

“Potential confounders of our observations, should they have differed systematically between HCWs on red and green wards and/or have changed over the course of the study, include: …… (b) Rates of vaccination of HCWs on red and green wards; however, the proportion of high-risk HCWs at CUHNFT offered vaccination prior to 08/01/21 was very low, and the study period was ended on 17/01/21 (before any substantial impact of vaccination was expected).”

4. Given the sophisticated mathematical modelling and statistical analysis, please address the danger of trying to overanalyse the intervention with relatively small numbers.

We do not believe that our mathematical model is particularly complex. In our model HCWs can be infected either via exposure to the virus while they are on a ward (which we describe in terms of a ward-specific infection risk), or they can be infected via exposure to the virus while they are not on a ward (which we describe as a non-ward-specific infection risk). We assume that ward-specific risks are constant within each type of ward, with the exception of a potential change following the change in PPE in red wards, and that non-ward-specific risks are constant across all types of ward. We are not certain that the questions addressed by our study could be evaluated with a less complicated model.

A very valuable point as suggested by the reviewer is that relatively small numbers of cases were available for our study, which additionally considered data from a single hospital and during a specific period of the SARS-CoV-2 pandemic. The size of these confidence intervals for all of the estimated parameters from our study directly reflects the amount of data we were able to collect. Given more data, we could be more precise in our estimates. We have added a sentence to our Discussion clarifying the extent to which sample size constrains the precision of our estimates.

“Our maximum likelihood inference that FFP3 masks (in combination with other PPE and infection control measures) provide 100% protection against ward-based infection should therefore be treated with caution; the large confidence intervals calculated for parameters in our model reflect the limited amount of data available.”

5. With this in mind it would be nice to see data from critical care ward staff, if they were using FFP3 throughout even though the numbers are small.

We have added data from critical care ward staff to Supplementary File 1. As anticipated by the reviewer, numbers were small. For completeness, we produced an extended version of our model in which critical care wards were included as a separate category to red and green wards (Figure 3 – supplemental figure 2). This did not substantially change the outcome with respect to the effect of introducing FFP3 respirators, which is the central focus of our model.

6. We would recommend to pay some attention in the discussion to the comments from Andrew Goddard from the RCP bulletin.

Very many thanks for this suggestion. We have already communicated with Andrew Goddard about his comments in the RCP bulletin. Of note, Dr. Goddard accepted all of the points we made, stating “I accept all your points”. As the reviewer suggests, some useful additions to the paper can be gleaned from our communications with Dr Goddard, and we note changes to the manuscript that have been introduced below:

– “More regular routine testing of ‘green’ areas appears to have started when the change was introduced.” Staff on green wards were asked to test weekly over the 11 weeks of the study. From 22/12/20 we asked staff on red wards and those green wards with more vulnerable patients (transplant, oncology) to test twice weekly.” This is included in the Methods section: “From 22/12/20, twice-weekly swabbing was offered on red wards and on wards where the most vulnerable patients were cared for (for example, transplant and oncology patients).”

– “The authors state that vaccination didn’t start in the relevant healthcare workers until after the intervention.” We set up our vaccination clinic to start vaccinating staff in early December but were required to vaccinate the over 80s first and so did not start the staff vaccination programme until 08/01/21. We gave spare vaccines to staff during December and early January from a list of those identified to be vulnerable (who were restricted from working on red wards) and the ED (whose staff were not included in the study). This has been addressed in the Methods section in a paragraph about vaccination, starting “A programme of SARS-CoV-2 vaccination using the BNT162b2 COVID-19 vaccine…”

– “Infection rates in healthcare workers in the second wave were 25% of those seen in the first wave due to previous infection, general improvement in infection control and vaccination.” Our first wave infection rates were possibly lower than some, with positive serology July/August 2020 being 7.2% (9·47% designated Covid-19 areas versus 6·16% in all other staff). Whilst this may have been a confounding factor I think it would have a similar effect on red and green ward staff and so not impact substantially on the findings. However, it is another issue for us to address. We have addressed this point by adding the following to the discussion “Overall seropositvity revealed by testing in July and August 2020 was 7.2% (9·47% amongst staff from red wards versus 6·16% amongst all other staff)”

We have added a further note to our discussion (below), highlighting that the inference of 100% protection from FFP3 respirators should be treated with caution in light of the small numbers in our study. While our study suggests an advantage being gained from FFP3 masks we are keen that they should not be regarded as ‘bullet-proof vests’.

“Our maximum likelihood inference that FFP3 masks (in combination with other PPE and infection control measures) provide 100% protection against ward-based infection should therefore be treated with caution; the large confidence intervals calculated for parameters in our model reflect the limited amount of data available.”

7. Although the manuscript states that a similar number of HCWs worked on each ward, it would be helpful to see the results displayed as a proportion of staff on red/green wards that became infected, or to model as individual exposure hours rather than ward status but I understand this might be difficult with the small sample size.

We have converted the denominator of our statistics to consider representative numbers of HCW days, rather than ward days, based upon rostering information for each of the wards in our study. We do not have readily available information about how many individual HCWs worked on each ward so as to calculate the proportion of staff who became infected. However, the change we have made leads to an estimated risk of infection per HCW day, rather than simply per ward day. Some of the derived numerical values have changed slightly as a consequence of this adjustment, for example the ratio of red/green risk.

8. It would be helpful for the number of red wards (or total exposure time) and C_i+1 to be plotted to give a visual representation of the correlation.

We thank the reviewer for this suggestion. We found a positive and significant correlation between the number of hours worked on red wards and community incidence, and have shown this in a new Figure 1S2.

9. As ward colour in the model is determined exclusively by the presence of infected patients and there is a possibility that one infected HCW on a green ward could lead to an outbreak within the HCW population. While this would not affect the result that changing the type of RPE used reduced the infections per ward day of HCWs on red wards, ignoring the potential for HCWs to seed outbreaks between ward staff could result in a higher proportion of cases being attributed to the community than actually occurred, and I would consider adding a third component of 'positive result in ward staff' to the model.

This is an important point. The ward colour in the model is determined by the theoretical presence of infected patients, since infected patients can still be present on green wards if undetected as such (Illingworth, Hamilton et al., eLife 2021). Furthermore, the models include the possibility of HCW-to-HCW infection, as they do not distinguish the origin of infections (patients vs HCWs). To address this point, we have added further clarity to the meaning of ‘ward-specific risk’ and ‘non-ward-specific’, or ‘community risk’. HCW-to-HCW infection can occur in two settings: (a) whilst working on the wards. This type of infection would be part of ‘ward-based risk’, and would be susceptible to modification by a change in RPE. This is because throughout the study, FFP3 masks were worn on a sessional basis, being changed at lunchtime and at the end of the day. (b) outside wards. Here, HCWs from both ward settings would experience the same risk of infecting each other, and this would be considered to be part of a ‘community risk’. Of note, one caveat is that HCWs on red wards more frequently ate lunch within the facilities available on the wards, to avoid the need to completely change out of the surgical scrubs worn as part of their PPE. If HCW-to-HCW infections acquired during these lunch breaks accounted for a significant part of the red ‘ward-based risk’, this would tend to act to reduce the effect of the change in RPE. A more nuanced understanding of ‘community’ risks could be gained with more information about patterns of mixing between HCWs during the period of the study, but this information was not available to us.

We have added the following to address these points:

Results:

“To further quantify the risk of infection for HCWs working on red and green wards, we generated a simple mathematical model. According to this model, the total risk of infection is divided into a risk from community-based exposure, and a risk from direct, ward-based exposure to patients *(ward-specific risk)*. The risk from direct exposure on red wards was allowed to vary upon the introduction of FFP3 respirators, and was fitted to a maximum likelihood model.”

Discussion:

“Whilst using FRSMs, the majority of infections among HCWs working on red wards could be attributed to direct exposure to patients with COVID-19. In contrast, as community incidence rose, the majority of infections among HCWs working on green wards were attributed by our model to community-based effects. After the change in RPE, cases attributed to ward-based exposure fell significantly, with FFP3 respirators providing an inferred 52-100% (most likely 100%) reduction in the risk of ward-based infection from patients with COVID-19.”

“Potential confounders of our observations, should they have differed systematically between HCWs on red and green wards and/or have changed over the course of the study, include:

(f) Differences in patterns of HCW behaviour on red and green wards, including mixing between HCWs from different areas. For example, staff working on green wards may have been more likely to leave the ward for lunch than staff working on red wards. Whilst such differences could in theory have contributed to the greater risk of HCW infection on red wards, they are unlikely to have changed systematically with the change in RPE. In addition, if mixing between HCWs from different areas led to an increased rate of infection, it would have tended to minimise any difference in ward-specific risk of infection.”

10. A sensitivity analysis shifting the 5-day cut off for before/after change in RPE and ward classification would also be a nice addition.

We have carried out a sensitivity analysis, shifting the cutoff to values between three and seven days (new Figure 3 —figure supplement 1). Altering the cutoff did not alter the maximum likelihood inference (100%) for the effect of FFP3 respirators, although the lower bound of the effect changed to between 30% and 72% with cutoffs between 3 and 7 days.

11. As staff working across wards are excluded, it is not clear if the full numbers of staff present on the red/green wards are included in estimations of both the number of cases and number of infectious sources (do the staff, and the non-ward based staff, interact similarly with patients and staff from both red and green wards?).

Many thanks for this point. Calculations of staff numbers on red and green wards were performed in terms of the numbers of nurses and health care assistants rostered to each ward on each day, based upon weekday and weekend patterns of working on each ward in the study. This provides a proxy for the total number of HCWs on any given ward. Differences in the interactions between HCWs and patients on red and green wards will contribute to differences in the ward-specific risk. Differences in the interactions between HCWs on red and green wards outside of the ward could not be measured.

We have added the following to address these points:

Methods

“The CUHNFT electronic rostering system recorded to which ward(s) individual nurses and healthcare assistants (HCAs) were allocated. Although this does not encompass 100% of ward staff, the data can be used to indicate relative ward size.”

“The number of ‘HCW days’ for each week of the study was calculated for each category of ward. Rostering information was used to identify the number of nurses and HCAs regularly assigned to each ward on each of the seven days of the week. Data describing the number of other staff on each ward was not available, but was assumed to be proportional to the number of rostered health care workers, calculations being performed in terms of nurse and health care assistant numbers.”

“In order to quantify the effect of the change in RPE upon cases in red wards, a mathematical model was developed, considering the numbers of cases observed among HCWs as arising from a combination of ward-specific infection risks, which relate directly to working on a red or green ward, and non-ward-specific risks, which include infections arising from the community.”

Discussion, ‘limitations’ section:

“Potential confounders of our observations, should they have differed systematically between HCWs on red and green wards and/or have changed over the course of the study, include:

(f) Differences in patterns of HCW behaviour on red and green wards, including mixing between HCWs from different areas. For example, staff working on green wards may have been more likely to leave the ward for lunch than staff working on red wards. Whilst such differences could in theory have contributed to the greater risk of HCW infection on red wards, they are unlikely to have changed systematically with the change in RPE. In addition, if mixing between HCWs from different areas led to an increased rate of infection, it would have tended to minimise any difference in ward-specific risk of infection.”

12. It would be interesting to see the results from the critical care wards that are currently excluded. If these wards used FFP3 respirators throughout, (noting the small numbers) it would be very helpful to see the data for this period in these wards as a (albeit limited) control group.

Many thanks for this suggestion; we have now addressed this in point 5 above.